# Feeding Spray-Dried Porcine Plasma to Pigs Reduces African Swine Fever Virus Load in Infected Pigs and Delays Virus Transmission—Study 1

**DOI:** 10.3390/vaccines11040824

**Published:** 2023-04-10

**Authors:** Elena Blázquez, Joan Pujols, Fernando Rodríguez, Joaquim Segalés, Rosa Rosell, Joy Campbell, Javier Polo

**Affiliations:** 1IRTA, Centre de Recerca en Sanitat Animal (CReSA), 08193 Barcelona, Spain; elena.blazquez@irta.cat (E.B.); joan.pujols@irta.cat (J.P.); fernando.rodriguez@irta.cat (F.R.); 2APC Europe, S.L. 08403 Granollers, Spain; 3Unitat Mixta d’Investigació IRTA-UAB en Sanitat Animal, Centre de Recerca en Sanitat Animal (CReSA), Campus de la Universitat Autònoma de Barcelona (UAB), 08193 Barcelona, Spain; joaquim.segales@irta.cat (J.S.); rosa.rosell@irta.cat (R.R.); 4WOAH Collaborating Centre for Emerging and Re-Emerging Pig Diseases in Europe, IRTA-CReSA, 08193 Barcelona, Spain; 5Departament de Sanitat i Anatomia Animals, Facultat de Veterinària, Campus de la Universitat Autònoma de Barcelona (UAB), 08193 Barcelona, Spain; 6Departament d’Acció Climàtica, Alimentació i Agenda Rural, Generalitat de Catalunya, 08193 Barcelona, Spain; 7APC LLC, Ankeny, IA 50021, USA; joy.campbell@apcproteins.com

**Keywords:** African swine fever, ASFV, spray-dried porcine plasma, challenge, nutritional intervention

## Abstract

The objective of this study was to evaluate the potential benefits of feeding spray-dried porcine plasma (SDPP) to pigs infected with African swine fever virus (ASFV). Two groups of twelve weaned pigs each were fed with CONVENTIONAL or 8% SDPP enriched diets. Two pigs (trojans)/group) were injected intramuscularly with the pandemic ASFV (Georgia 2007/01) and comingled with the rest of the pigs (1:5 trojan:naïve ratio) to simulate a natural route of transmission. Trojans developed ASF and died within the first week after inoculation, but contact pigs did not develop ASF, viremia, or seroconversion. Therefore, three more trojans per group were introduced to optimize the ASFV transmission (1:2 trojan:naïve ratio). Blood, nasal, and rectal swabs were weekly harvested, and at end of the study ASFV-target organs collected. After the second exposure, rectal temperature of conventionally fed contact pigs increased >40.5 °C while fever was delayed in the SDPP contact pigs. Additionally, PCR Ct values in blood, secretions, and tissue samples were significantly lower (*p* < 0.05) for CONVENTIONAL compared to SDPP contact pigs. Under these study conditions, contact exposed pigs fed SDPP had delayed ASFV transmission and reduced virus load, likely by enhanced specific T-cell priming after the first ASFV-exposure.

## 1. Introduction

African swine fever virus (ASFV) is an enveloped DNA virus of the Asfarviridae family that infects domestic pigs and wild boar of all ages causing African swine fever (ASF), a disease that must be reported to the World Organization for Animal Health (WOAH, formerly OIE).

In January 2014, Lithuania made the first notification of acute ASF cases in wild boar in the European Union (EU), and Poland followed in February 2014. In June and September 2014, Latvia and Estonia also reported ASFV. To date, Ukraine, Romania, Estonia, Hungary, Bulgaria, Czech Republic, Slovakia, Belgium, Greece, Germany, and, recently, Italy, have reported cases of ASF in wild boar and some countries in domestic pigs. All epidemiological data indicate that the EU has undergone repeated introduction of ASFV from the Eastern neighboring infected countries. The EU Reference Laboratory confirmed, through genetic studies that there was a 100% homology with the genotype II virus that entered Georgia in 2007, imported from Western Africa [1]. Furthermore, from August 2018, ASFV has spread in most pig producing countries in Asia, including China, Vietnam, South and North Korea, Philippines, Laos, Myanmar, Mongolia, Cambodia, Timor-Leste, Indonesia, Malaysia, India, Papua New Guinea, and Thailand. In addition, genotype II ASFV has recently spread to the Caribbean, and new cases in Haiti and Dominican Republic have been reported in wild and domestic pigs [2].

However, the pattern spread and severity of ASF outbreak in domestic pigs in different regions varies significantly, ranging from up to 70% reduction of total swine herd in China, Vietnam, Philippines, and other Asia countries to limited spot outbreaks in EU countries and South Korea. Biosecurity could be the main factor influencing the spread and outcome of the disease, as countries with less severe outbreaks had better biosecurity programs. The health and immune status of the herd may also perform a key role. Pietschmann et al. [3] reported that immune compromised runt and gastrointestinal challenged unhealthy pigs are more susceptible for ASF infection. Radulovic et al. [4] reported that domestic pigs with a very high hygienic and health status experienced a milder and shorter form of the illness followed by complete recovery when inoculated with an attenuated field ASFV strain. Therefore, improving health and hygienic status of the herd may be a key for ASF prevention and control.

Spray-dried plasma (SDP) is a dry functional feed ingredient that contains a diverse mixture of many functional components, such as immunoglobulins, albumin, growth factors, biologically active peptides, transferrin, and other molecules, that have biological activity independent of their nutritional value [5,6,7,8,9,10,11]. SDP either from porcine (SDPP) or bovine (SDBP) origin are extensively used in pig starter diets and consistently provide improvements in growth performance, feed efficiency, and animal survival, especially under stressful conditions, such as a pathogen challenge [6,12]. The beneficial effects of SDP are related to its mode of action that supports an efficient immune system response [8,9,13]. In addition, feeding SDPP has been shown to promote a more favorable gut microflora [14,15,16]. Thus, SDPP helps improve the integrity of the intestinal barrier by modulating the functional and structural properties of the intestinal mucosa in pigs and other animal models [17]. In addition, dietary SDPP can favorably affect intestinal morphology and immune cell subsets of gut tissues and blood in weaned pigs [18]. Furthermore, SDPP exerts a systemic effect, as demonstrated by the reduction of lung inflammation and the presence of monocytes and cells involved in the inflammatory events under an acute pulmonary inflammation model with LPS in mice [19,20]. Moreover, in an epidemiologic study conducted in Manitoba, Canadian researchers found that farms with less pulmonary lesions associated with the presence of porcine reproductive and respiratory syndrome virus (PRRSV), porcine circovirus 2 (PCV-2), and Mycoplasma hyopneumoniae were well correlated with supplementation of SDPP in their feed [21].

Good intestinal health translates into a more efficient immune system and overall better growth performance [22,23]. Following this principle, an improved intestinal barrier function through gut microbiota has been demonstrated to influence ASFV susceptibility. Fecal transplantation from warthogs (African pigs resilient to ASF) to domestic pigs improved mucosal immunity and protected pigs against experimental challenge with a live attenuated ASFV strain [24]. Although mechanisms involved in ASFV protection are not fully understood, ASFV specific antibodies [25] and cytotoxic CD8 T-cell (CTL) responses [26], together with an appropriate innate immune response [27], have been demonstrated to perform an important role. The activation of Th1-like responses, including specific CD8 T-cells, is a strategy currently pursued to obtain efficient vaccines against ASFV [28]. The fact that dietary SDPP can protect mucosal integrity and also promote optimal immune responses, including Th1-like responses and CTL induction [29,30], prompted us to hypothesize that SDPP could be used as a potential nutritional intervention against ASFV transmission and progression of the infection.

Therefore, the objective of this study was to assess if supplementing SDPP in diets of healthy pigs reduced ASF virus load of infected animal after an experimental ASFV pig-to-pig contact challenge.

## 2. Materials and Methods

### 2.1. Ethical Statement

The study was approved by the committee of ethics and welfare “Comitè d’Experimentació Animal de la Generalitat de Catalunya” with the protocol approval number CEA-OH/11025/1. For this study, 24 Landrace × Large White male pigs (5 weeks of age) were obtained from a commercial farm with high sanitary status.

The clinical state of the animals and the end-point criteria was evaluated by scoring the ASF-compatible clinical signs following a previously reported guide [31]. A score from 0 to 5 according to severity was applied as follows: 0: no clinical signs, 1: mild pyrexia (39.6–40.0 °C), 2: mild pyrexia (39.6–40.0 °C) and mild clinical signs (skin, digestive), 3: moderate pyrexia (40.0–40.5 °C) and mild-moderate clinical signs (distal ear spots, mild limp, lying down, but remaining alert), 4: moderate-high pyrexia (40.5–41 °C) and moderate clinical signs (remains dormant, only stands up when touched, hesitant step, subcutaneous bleeding <10%, diarrhea, mild tremors), and 5: pyrexia higher than 41 °C and moderate-severe clinical signs (generalized subcutaneous bleeding, ataxia, spasticity, clouding, prostration, and bloody diarrhea).

### 2.2. Study Design

The 24 pigs were divided randomly into 2 groups of 12 pigs each and were placed in 2 separate rooms at the IRTA-CReSA biosecurity level 3 animal facility. Each group of 12 pigs represented an experimental treatment group: CONVENTIONAL: ASFV challenged pigs fed a conventional diet with soy protein concentrate, and SDPP: ASFV challenged pigs fed a diet with SDPP. The SDPP used in this study was randomly selected from a manufactured lot of AP920 produced by APC Europe S.L.U. (Granollers, Spain). The CONVENTIONAL group diet was formulated with soy protein concentrate (Soycomil^®^ P, ADM Animal Nutrition™, Quincy, IL, USA) at 100.9 g/kg and the SDPP group was fed a diet formulated with SDPP at 80 g/kg, which entirely replaced Soycomil with small adjustments in other ingredients and synthetic amino acids. Both diets were iso-energy (3340 kcal ME/kg) and iso-nutrient balanced (Lysine SID 12.50 g/kg feed). The experimental diets (Table 1) were prepared in mash form and offered ad libitum during the adaptation period and the 31-day study. The rooms contain slated floor and the environmental conditions for both rooms were set at 22 ± 2 °C and relative humidity of 60 ± 5%. The air renewal was established to be 12 times/hour. The feed was provided each morning between 7:30–9:30 a.m. All procedures involving clinical data taking, necropsies, and laboratory analyses were performed in a blind fashion in regards the treatments.

After a brief adaptation period to the diets and facility, on day 4 of the experiment (d-2) two randomly selected animals (for use as trojan pigs) per room were separated from the other pigs (10 contact pigs per room) by placing them in two subdivisions to avoid direct contact with the other pigs. Immediately thereafter, the two trojans per room were intramuscularly (IM) injected with a lethal dose of 10^3^ genome equivalent copies (GEC) of the Georgia 2007/1-ASFV strain. Two days after virus inoculation, the trojan and contact pigs in each room were re-grouped to begin the contact exposure (day 0 post-exposure, d0pe). The ratio of trojan to naïve contact pigs was 1:5, mimicking a potential field scenario of slow ASFV transmission. As expected, trojans developed ASF signs by day 4 after inoculation and were sacrificed for humane reasons by day 6 after inoculation when showing acute ASF clinical signs.

However, none of contact pigs in either group developed fever, ASF specific signs, or viremia after 17 days of observation (d17pe). In view of these results, a modification of the protocol was obtained from the ethical committee to increase the infection pressure by providing a second exposure of ASFV infected trojan pigs to contact pigs. Accordingly, three pigs from each group were chosen randomly and inoculated IM with 10^3^ GEC (acting as trojan seeders) while remaining inside the same pen with the six remaining contact pigs to have a 1:2 (trojan to contact pig) ratio. To maintain this optimal contact challenge ratio [32], one pig per each contact group was excluded from the study before the second ASFV-exposure. The study ended at d29pe (day 12 post-second exposure, d12pse), following the originally approved protocol (Figure 1).

During the whole study animals were daily observed for ASF-compatible clinical signs (including rectal temperature) and samples of blood (EDTA 15 mL tubes) were taken from all animals at −2 days post-contact immediately before IM injection to the 2 trojans, d2pe, d5pe, d12pe, d19pe (d2pse), and d26pe (d9pse). In addition, nasal and rectal swabs were collected at d2pe, d5pe, d9pe, d12pe, d16pe, d19pe (d2pse), d23pe (d6pse), and d26pe (d9pse). Necropsies were done on d29pe (d12pse), following the schedule in Figure 1. All samples were stored at −75 °C until processed. Blood, nasal, and rectal swabs were analyzed by real time quantitative PCR (qRT-PCR) to detect viremia. At necropsy, lesions were registered, and samples of spleen, tonsil, gastro-hepatic node, submaxillary node, and retropharyngeal node were taken and analyzed for virus detection by qRT-PCR.

ELISPOT analysis [33] was conducted in frozen cells where viability was >90% after thawing from blood obtained with EDTA at d9pe, d23pe (d6pse) and d29pe (d12pse).

### 2.3. Laboratory Analyses

Samples of ASFV inoculum were analyzed by the qRT-PCR described by [34]. Blood samples were taken in tubes with EDTA on specified days. DNA extraction was done using the Indimag Pathogen Kit (Indical Biosciences, Leipzig, Germany). Viremia was determined by qRT-PCR analysis using the primers described by Fernández-Pinero et al. [35], and the probe ASF-VP72P1 described in the current OIE ASF chapter (Terrestrial Manual OIE, Section 3.9, Chapter 3.9.1 African Swine Fever Virus pages 1–18) with the following modification in the thermoprofile made by the Spanish National Reference Laboratory for ASF: 10 min at 95 °C, 5 cycles 1 min at 95 °C + 30 seg at 60 °C, 40 cycles 10 min at 95 °C + 30 seg at 60 °C with fluorescence acquisition in the FAM channel at the end of each PCR cycle. According to these amplification settings results were considered as positive when Ct values were ≤30, inconclusive Ct values between 30 to 35, and Ct > 35 were considered negative.

Seroconversion was determined by ELISA (Ingezim PPA COMPAC, INGENASA; Madrid, Spain) with O.D. ≤ 0.061 as negative. Nasal and rectal swabs were analyzed for the presence of ASFV genome using the same procedures previously mentioned. Once the nasal and rectal swabs arrived at the laboratory, the end of the swab was cut and placed in a tube with 1 mL of PBS. The tubes were stored at −75 °C until DNA extraction and analysis by qRT-PCR.

At necropsy, samples of tonsil, spleen, and retropharyngeal, submaxillary, and gastro-hepatic lymph nodes were taken for subsequent qRT-PCR analysis. For each sample, 0.1 g of tissue was diluted 1:10 and homogenized using sterile PBS and TyssueLyser II (Qiagen, Hilden, Germany). DNA extraction and qRT-PCR were completed as detailed above.

Correlation Ct vs. Log10 HAD50/mL of the different analyzed tissues was carried out using ASFV Georgia 2007 viral stock at 6.99 Log10HAD50/mL as a standard. qRT-PCR were carried out as detailed above (Appendix A).

Peripheral blood mononuclear cells (PBMCs) were purified from EDTA blood samples by density-gradient centrifugation with Histopaque 1077 (Sigma-Aldrich, St. Louis MI, USA). Cells were frozen at −80 °C, but the viability was tested to be ≥90% after thawing and before conducting the analysis. Commercial antibody Porcine IFN-γ P2G10 and biotin P2C11, from BD Biosciences (Pharmingen, San Diego, CA, USA) at 5 µg/mL was used to quantify by ELISPOT assay the number of IFNγ secreting cells. PBMC were stimulated against ASF Georgia/01 strain virus at a MOI 0.2 to measure specific T-cell responses by ELISPOT in real time with fresh cells [35]. Phytohemagglutinin (PHA; Roche Diagnostics, Barcelona, Spain) was used as a positive control and the mock was carried out using only cells and RPMI medium as a negative control. Plates were revealed using detection antibody from BD Pharmingen (BD Biosciences Pharmingen, San Diego, CA, USA), Strep-HRP from Life Technologies (South San Francisco, CA, USA) and TMB substrate for ELISPOT assay (MABTECH, Stockholm, Sweden). ELISPOTs were read under a magnifying glass. The value obtained for each animal was obtained subtracting the corresponding values of mock-stimulated cells.

### 2.4. Statistical Analysis

Data were analyzed as a completely randomized design using the GLM procedures of SAS (SAS Inst., Inc., Cary, NC, USA). An analysis of variance was conducted to detect differences among treatments. The independent variable was treatment. Dependent variables were body temperatures, blood, nasal, and rectal swab, and tissue Ct values. The LSMEANS procedure was used to calculate the mean values by treatment. If treatment effects were detected, least squares means were separated using the PDIFF option in SAS. Pigs were considered the experimental unit. Means are considered significantly different if *p* < 0.05 while trends are reported as *p* = 0.05 to 0.10.

## 3. Results

### 3.1. First Exposure

After the first exposure to ASFV-inoculated trojans (1:5 ratio of trojans:contact pigs), none of the contact pigs developed temperatures above 40.4 °C, had ASF compatible signs or viremia after 16 days of monitoring (Appendix A). One CONVENTIONAL contact pig on d2pe, and 5 CONVENTIONAL and 3 SDPP contact pigs on d5pe displayed an ASFV qRT-PCR positive result in the nasal swab, but thereafter to d16pe, all nasal swab results were negative. All blood and rectal swab samples for contact pigs were negative. Despite the nasal swab results for contact pigs, this did not result in a disseminated infection (Appendix A), even though trojan pigs in both groups had developed high fever and were euthanized d4pe per protocol requirements.

### 3.2. Second Exposure

Second exposure: Conversely, after the second contact exposure (1:2 ratio of trojans:contact pigs), contact pigs from the CONVENTIONAL group became infected by ASFV, following the expected kinetics [36]. Thus, all six contact pigs from the CONVENTIONAL group developed acute ASF-compatible clinical signs, showing high fever (rectal temperature > 41 °C) by d29pe (d12pse) with one contact pig (#15) having >41 °C starting at d26pe (d9pse; Figure 2A). Interestingly, all contact pigs from the SDPP group had a delay in the onset of ASF clinical signs with only two of the six pigs (#4 and 7) having rectal temperatures >41 °C the last day of the study (Figure 2B; Appendix A).

Average body temperature was higher (*p* < 0.05) for CONVENTIONAL contact pigs on d23pe (d6pse), and d26pe to d29pe (d9pse to d12pse) compared to SDPP contact pigs (Figure 3).

All three trojan pigs in the CONVENTIONAL group died between 4 to 7 days post-inoculation. However, two of the three trojan pigs in the SDPP group survived until the last day of the study with RT below 41 °C, even though these pigs displayed severe signs of disease. Therefore, the exposure time of contact pigs to trojans displaying fever was longer in the SDPP group than in the CONVENTIONAL group due to the longer survival rate of the trojan pigs in the SDPP group.

Viremia was detected in both groups at d29pe (d12pse). ASFV qPCR positive nasal swab was presented in all pigs for both groups from d23pe (d6pse) probably confirming the contact of these pigs with the second group of trojans. ASFV qPCR positive rectal swab was found in one pigs of the non-treated group on d23pe although at very low level (Ct 33); in contrast, the presence of ASFV virus in feces of pigs in the SDPP group was not found until d26pe (d9pse). Figure 4; Appendix A.

At necropsy, trojans from SDPP group showed lower load of virus genome in their tissues compared with the trojans in CONVENTIONAL group.

Despite the longer exposure time of contact pigs in the SDPP group to the trojans (two-thirds survived), contact pigs in the SDPP group had delayed onset of clinical sign, lower viremia load, ASFV shedding, and ASFV genome copies or Log10HAD50/mL in every single tissue analyzed on d29pe (d12pse), compared to contact pigs in the CONVENTIONAL group, which may indicate reduced rate of in-pen transmission (Figure 5; Appendix A).

Viremia was found on d29pe (d12pse) in all animals except for one pig from the SDPP group (#5) and the average Ct values using the standardized qRT-PCR approved by the OIE in the SDPP group was numerically higher (indicating less ASFV genome copies) than in the CONVENTIONAL group. On d29pe (d12pse), nasal swabs from all animals were qPCR positive for ASFV, but again, the average Ct values in the SDPP group was numerically higher than the CONVENTIONAL group. The rectal swab average Ct values were low and almost identical for both groups with one and two pigs showing no detectable ASFV genome in the SDPP and CONVENTIONAL groups, respectively (Figure 4). Finally, submaxillary lymph nodes and tonsil tissue for all contact animals in both groups were qPCR positive, except for animal #3 of the SDPP group that was negative (Ct value > 35). In gastro-hepatic and submaxillary lymph nodes there was a tendency (*p* < 0.10) to have less average amount of virus genome copies (higher Ct values) for animals in the SDPP to the CONVENTIONAL group. In addition, pigs in the SDPP group had lower (*p* < 0.05) ASFV load in tonsil and retropharyngeal lymph nodes.

To understand the mechanisms explaining the differential outcome of both ASF clinical sign evolution and virus transmission, the kinetics of both the specific antibody and cellular responses were followed. None of the contact pigs from either group developed detectable anti-ASFV antibodies after the first ASFV contact challenge, which is consistent with the absence of detectable viremia. Even after the second exposure to ASFV inoculated trojans, contact pigs did not have time to develop significant amounts of anti-ASFV antibodies and by d12pse remained below the threshold limit of the technique (Appendix A). In clear contrast with the lack of antibody responses, ASFV-responding IFN-γ secreting cells were detectable by ELISPOT as soon as 9 days after first ASFV encounter (d9pe); thus, 8 of 9 contact pigs in the SDPP group showed low, albeit detectable ASFV-responding cells, while 2 of 7 contact pigs in the CONVENTIONAL group had a detectable response. On d23pe of the experiment (d6pse), most pigs in both groups developed a detectable cellular response against ASFV measured by ELISPOT (Figure 6; Appendix A).

## 4. Discussion

Despite the dramatic consequences that ASF represent for the swine industry and for global trading, ASFV cannot be considered as a highly transmissible virus, at least compared with other pig viruses of obligatory declaration to the WOAH [37], such as classical swine fever virus (CSFV) and foot and mouth disease virus (FMDV). In agreement with this observation, naïve pigs fed daily for 14 consecutive days with feed blended with liquid porcine plasma inoculated with 10^4.3^ or 10^5.0^ TCID50/g ASFV remained ASF-free [38]. The fact that ASFV recovered from the feed remained infectious when administered intra-muscularly or intra-gastric with a probe, confirmed that infection with ASFV through feed consumption is not as efficient as for other pathogens. Other studies feeding commercial inoculated ASFV contaminated feed not containing plasma resulted in infection of the naïve pigs [39,40,41]). These results suggest that unprocessed liquid porcine plasma or SDPP may contain inherent properties that diminish the infectious capacity of ASFV and may contribute to the survival of two of the three SDPP trojans until end of the present study.

This limited capability of ASFV transmission might explain the lack of evident viral transmission observed between trojan and contact pigs at a ratio of 1:5 (infected vs. non-infected animal ratio). However, during the first exposure the contact pigs were probably exposed to the virus at low levels for a short time, which was adequate to trigger the induction of a mild ASFV-responding cellular response but was not sufficient to induce detectable ASFV-specific antibodies or viremia at the times tested. This finding was definitively confirmed after showing that some contact pigs in both groups had low amounts of ASFV genome detectable by qPCR in the nasal swab by d2pe and d5pe.

The trojans in contact ASFV challenge model used in this study are as reproducible as the direct IM ASFV inoculation model. This is a reliable challenge model [32] if the optimal ratio (1:2) of trojans versus contact pigs is kept. Moreover, we consider that this is a more natural model of infection compared to the IM model. The difference between both models (IM vs. in contact) is the kinetics of the infection. Thus, the onset of infection is faster for IM injected pigs with fever and ASFV being detectable from day four post-challenge. Conversely, in contact challenge pigs develop similar clinical signs, including fever, viremia, and shedding with 3 days of delay, reaching similar maximum virus titers and ASF compatible signs than those injected IM [32].

The presence of low levels of ASFV in the nostrils of contact pigs without causing viremia or ASF clinical signs might be further explained by the induction of efficient innate immune responses capable of controlling ASFV infection [27] at the site of entry. The fact that positive nasal swab samples were detected in a lower number of contact pigs in SDPP versus the CONVENTIONAL group, can be related with either less virus excretion from the trojan pigs fed SDPP diet during first exposure or related to our original hypothesis of innate response improvement by SDPP feeding [18,29]. These results fit with the fact that at later time points (d9pe), most contact pigs fed SDPP showed ASFV-responding IFN-γ secreting cells, compared with only two animals fed the CONVENTIONAL diet. The positive effect of SDPP on Th1-like memory responses was also confirmed by others [29,30], including the induction of cytotoxic CD8+ T cells [29,30,42]. Independently of this, as mentioned earlier, Th1-like responses and CD8+ T cells perform a key role in ASFV protection [19,21] and dietary SDPP in other challenge studies has demonstrated increases in the percentage of CD8+ T cells [29,30,42].

The potential presence of ASFV specific CD8+ T-cells in contact pigs fed SDPP after the first exposure with the trojan pigs, may partially explain the delay of the ASF onset observed in this group of pigs after the second ASFV encounter [26,28,32], but also the unique feature observed in two of the trojan pigs used for the second challenge that survived 12 days after IM ASFV challenge. Despite that average viremia and nasal shedding genome values were similar between both groups of trojan by day 6 post challenge (Appendix A), these two pigs (#8, #9) in the SDPP group continued secreting ASFV to the environment until the end of the experiment (d29pe or d12pse).

SDPP is a functional ingredient that is demonstrated to improve the integrity of the intestinal barrier, enrich the gut microbiota, and modulate the immune system of pigs and other animal species [13,14,15,16,17,22,23,43,44,45]. Improving integrity of intestinal barrier function may reduce the potential infectivity of ASFV and microbiota seems to have an important role for controlling ASFV infection [24,46]. In addition, SDPP supplemented in feed was shown to modulate the immune system and balance the ratio between activated and regulatory T cells [13,17], which are essential to protect against most pathogens, including ASFV [27]. As above mentioned, albeit little is known about the mechanisms involved in ASFV protection, innate and both memory B and T cells responses perform a role, most probably in a regulated way to avoid excessive inflammation. Diets with SDPP may contribute to this regulation by providing an optimal cytokine environment [13,17,18,44]. The molecular mechanisms explaining why the contact-exposed pigs fed SDPP showed lower virus load compared to conventional pigs have not been elucidated. We believe that the accelerated cellular response (IFN-γ secreting cells) detected in this group of pigs after the first ASFV suboptimal encounter, might partially explain the protection afforded against the second challenge, this time using the optimal ratio of trojans. Recently, a study conducted by Radulovic et al., [4] demonstrated that pigs can regulate immune activation more efficiently after vaccination with attenuated ASFV strains; these animals showed milder production of proinflammatory cytokines after virus inoculation, similar to the results in previous studies when animals are fed diets supplemented with SDPP [13,17,44].

In Asia-Pacific regions, new, less virulent strains and attenuated strains of ASFV compared with the original Georgia strain have appeared, probably due to genetic evolution the virus and the use of non-authorized vaccine prototypes [47,48]. Such scenarios complicate the epidemiological ASFV picture and control of ASF. Based on the results from this study using a virulent ASFV strain we speculate that healthier pigs fed diets with SDPP may improve resilience to ASFV strains because diets with SDPP improves the health status of pigs, especially under stressful conditions [12].

Although the length of time the donor pigs were in contact with the other animals was greater for the SDPP pigs, the data suggests viraemia and shedding was lower which may also contribute to explain the differences observed, rather than the feed itself. Further work should be performed in the future with a larger group of animals to confirm these observations. However, the accelerated specific T-cell responses observed in SDPP-fed animals upon the first encounter with the ASFV, the lower ASFV loads found in the same group after the second challenge, together with the beneficial effect that SPPP seems to exert upon experimental ASFV vaccination [49] (back-to-back submitted manuscript) would support the hypothesis that nutritional interventions may be useful to help ameliorating the impact of ASF in endemic regions.

## 5. Conclusions

In conclusion, feeding SDPP reduced virus load in the different target tissues and may delayed in pen transmission in pigs exposed to ASFV inoculated trojans. Therefore, supplementation of SDPP in feed may be a strategic nutritional intervention to improve the degree of protection against ASFV.

## Figures and Tables

**Figure 1 vaccines-11-00824-f001:**
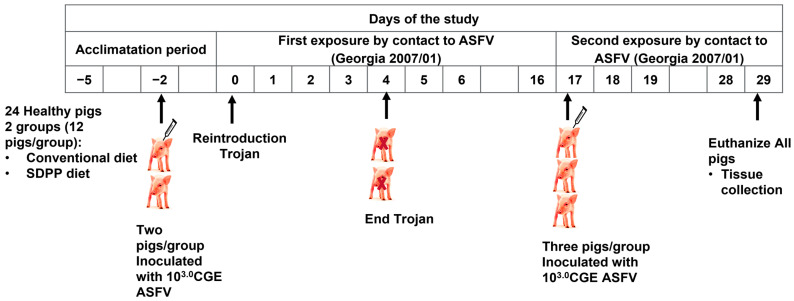
Schematic representation of the study design.

**Figure 2 vaccines-11-00824-f002:**
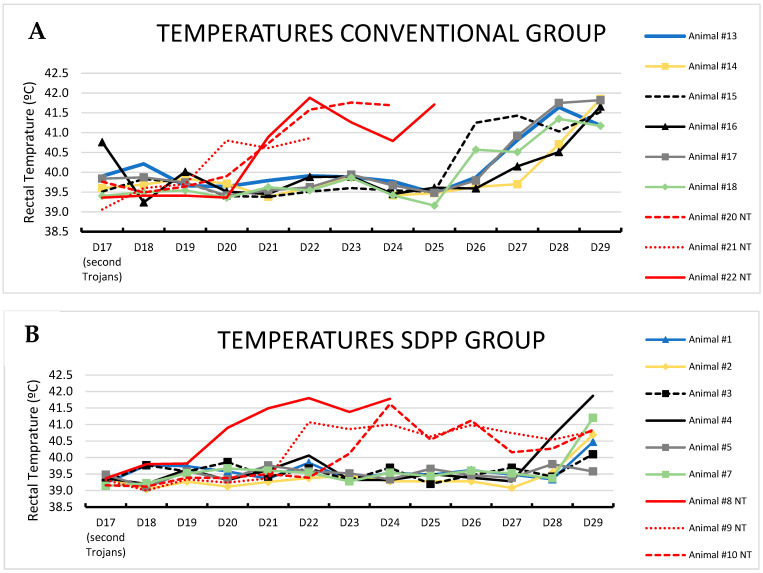
Rectal temperature over time after second ASFV exposure. (**A**): animals fed with CONVENTIONAL diet and (**B**): animals fed with spray dried porcine plasma (SDPP) diet. In red color are the trojans animals in each group.

**Figure 3 vaccines-11-00824-f003:**
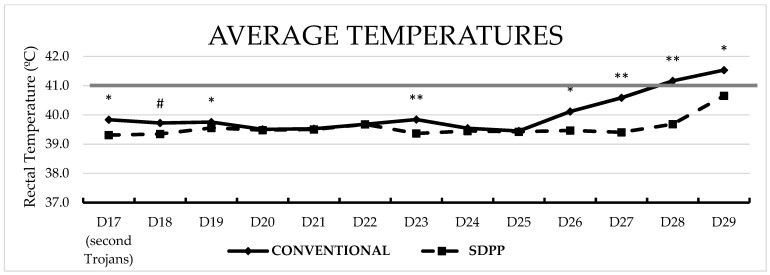
Average rectal temperature over time between treatments after second ASFV exposure. # = *p* < 0.1; * = *p* < 0.05; ** = *p* < 0.01 SDPP = spray-dried porcine plasma.

**Figure 4 vaccines-11-00824-f004:**
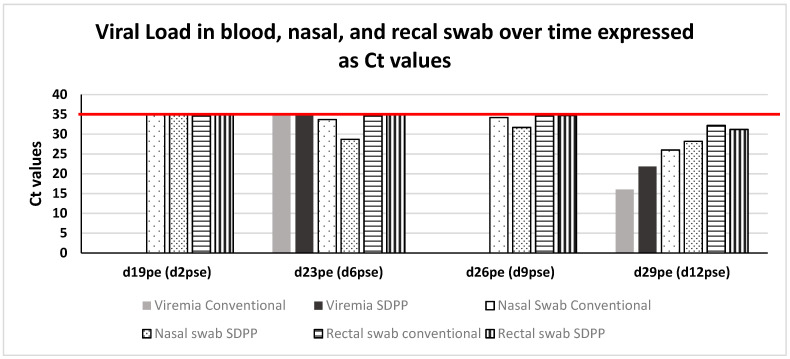
PCR results in blood, nasal and rectal swabs on different days during the study. Results expressed in Ct values/mL The red color bar indicated the limit of detection of qRT-PCR for positive samples.

**Figure 5 vaccines-11-00824-f005:**
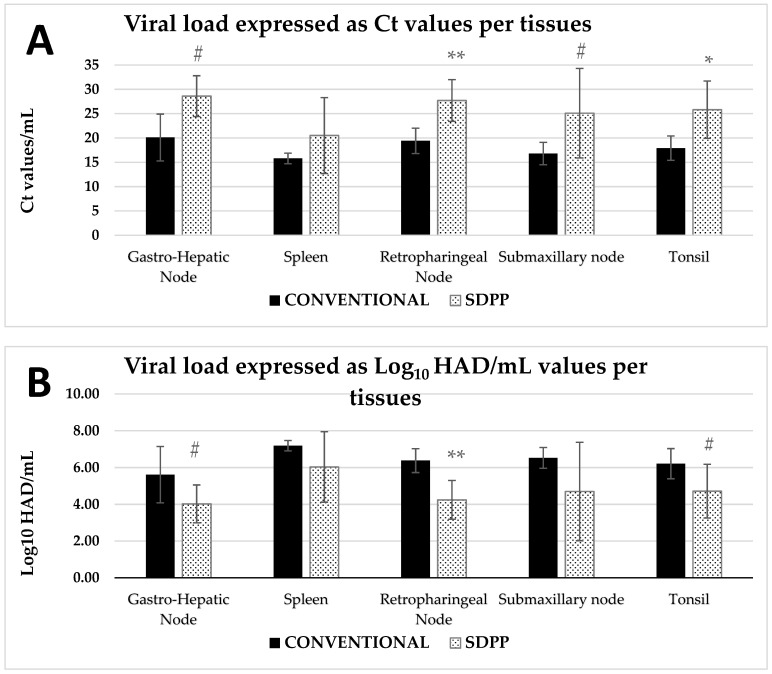
(**A**). Average qRT-PCR results of different tissues at d29pe (d12pse) of the study. Values expressed as Ct values/mL. (**B**). Average viral load expressed as Log_10_ HAD/mL in different tissues analyzed at d29pe (d12pse) of the study. **#** = *p* < 0.1; * = *p* < 0.05; ** = *p* < 0.01 SDPP = spray-dried porcine plasma.

**Figure 6 vaccines-11-00824-f006:**
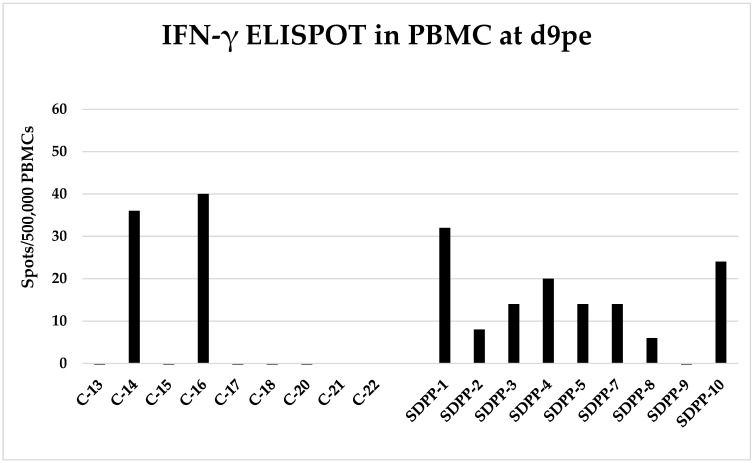
ASFV-specific IFN-γ secreting cells analyzed by ELISPOT at 9 days after first ASFV encounter (d9pe). PBMC from d9pe were stimulated in vitro with Georgia 2007/1 at MOI 0.2 and the number of ASFV-responding IFN-γ-secreting cells was quantified by ELISPOT. Values shown are individual values subtracting the corresponding values of mock-stimulated cells. C-#: Animals in CONVENTIONAL diet; SDPP-#: animals in spray-dried porcine plasma (SDPP) diet.

**Table 1 vaccines-11-00824-t001:** Composition and estimated nutritive composition of the basal experimental diet (g/kg feed).

Ingredients	Conventional Diet	SDPP Diet
Barley	298.7	313.3
Maize	295.7	303.4
Sweet milk whey	137.2	137.2
Soy protein concentrate	100.9	-
Spray-dried porcine plasma (SDPP)	-	80.0
Soybean meal (48% CP)	107.4	110.6
Animal fat	26.7	30.0
Dicalcium phosphate	14.4	16.1
Calcium carbonate	1.2	0.5
Salt	4.7	0.2
L-Lysine-HCl	4.2	2.3
L-Threonine	1.9	0.5
DL-Methionine	2.1	1.4
L-Tryptophan	0.7	0.2
L-Valine	0.3	-
Noxyfeed *	0.2	0.2
Vitamin-Mineral complex **	4.0	4.0
Nutritive composition		
Crude Protein	195.0	195.0
Crude Fiber	28.0	25.2
Fat	45.5	50.3
Ash	58.2	56.6
Energy (kcal ME/kg)	3340	3340
Total sodium	2.8	2.8
Total chlorine	6.7	4.5
Total calcium	7.5	7.5
Total phosphorous	6.7	7.4
Digestible phosphorous	3.8	3.8
Lysine (SID)	12.50	12.50
Threonine (SID)	8.13	8.13
Methionine (SID)	4.62	3.75
Met + Cys (SID)	7.38	7.78
Tryptophan (SID)	2.50	2.50
Isoleucine (SID)	7.36	6.75
Valine (SID)	8.50	9.26
Leucine (SID)	13.52	14.89
Phenylalanine (SID)	8.02	8.52
Phe + Tyr (SID)	13.41	14.60
Histidine (SID)	4.25	4.76

* ITPSA, Barcelona, Spain. Contains BHT+ propyl galate (56%) and citric acid (14%). ** Provides per kg feed: vitamin A (E-672) 10,000 UI; vitamin D3 (E-671) 2000 UI; vitamin E (alfa-tocopherol) 25 mg; vitamin B1 1.5 mg; vitamin B2 3.5 mg; vitamin B6 2.4 mg; vitamin B12 20 µg; vitamin K3 1.5 mg; calcium panthotenate 14 mg; nicotinic acid 20 mg; folic acid 0.5 mg; biotin 50 µg; Fe (E-1) (from FeSO_4_·H_2_O) 120 mg; I (E-2) (from Ca(I_2_O_3_)2) 0.75 mg; Cu (E-4) (from CuSO_4_·5H_2_O) 6 mg; Mn (E-5) (from MnO) 60 mg; Zn (E-6) (from ZnO) 110 mg; and Se (E-8) (from Na_2_SeO_3_) 0.37 mg.

## Data Availability

All data from this study is provided in the manuscript and Appendix A.

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
