# Peer review of "Feeding Spray-Dried Porcine Plasma to Pigs Reduces African Swine Fever Virus Load in Infected Pigs and Delays Virus Transmission—Study 1"

_vaccines, 2023, doi:10.3390/vaccines11040824_

Round 1
Reviewer 1 Report
The current study evaluated the potential benefits of feeding spray-dried 18
porcine plasma (SDPP) to pigs infected with African swine fever virus (ASFV). Two groups of weaned pigs were fed with either CONVENTIONAL or 8% SDPP enriched diets. Two trojan pigs infected with the pandemic ASFV (Georgia 2007/01) to the groups to simulate a natural route of transmission. Later on, three more trojans per group were introduced to optimize the ASFV transmission. Under these study conditions, contact exposed pigs fed SDPP had delayed ASFV transmission and reduced virus load when compared to the conventional ones.
1- it was not clear if the study was conducted blindly to avoid any bias?
2- From Table 1: SDPP is the not only difference in the two diets. How the authors can ensure that the observed data were due to SDPP and not the other ingredients?
3- line 173: what qRT-PCR stands for?
4- x axis lable is hiding bar's label. why these three dots in the label?
5- Line 270: although this is true, the data cannot actually reflect the actual situation. there is no house keeping gene normalization or accurate detection of genome copies.
6- do the authors think that ifecting animals by injection (not through contact) will show the same trend?
Author Response
The current study evaluated the potential benefits of feeding spray-dried porcine plasma (SDPP) to pigs infected with African swine fever virus (ASFV). Two groups of weaned pigs were fed with either Conventional or 8% SDPP enriched diets. Two trojan pigs infected with the pandemic ASFV (Georgia 2007/01) to the groups to simulate a natural route of transmission. Later on, three more trojans per group were introduced to optimize the ASFV transmission. Under these study conditions, contact exposed pigs fed SDPP had delayed ASFV transmission and reduced virus load when compared to the conventional ones.
- it was not clear if the study was conducted blindly to avoid any bias?
Yes, the study was conducted in a blinded fashion for the animal caretakers, pathologist and lab workers. Therefore, nobody getting data from the study knew the group they were working with. Thank you for pointing out this issue. We gave it for granted and should be corrected. A new sentence has been added in the revised manuscript in lines 134-136.
- From Table 1: SDPP is the not only difference in the two diets. How can the authors ensure that the observed data were due to SDPP and not the other ingredients?.
Thanks again for the observation. Yes, we do believe that the differences observed are due to SDPP. To clarify this issue, in the new version of the manuscript, we have added “and the SDPP group was fed a diet formulated with SDPP at 80 g/kg, which entirely replaced Soycomil with small adjustments in other ingredients and synthetic amino acids”. The objective of this study was to provide iso-nutritive diets for both groups with similar energy and balance iso-nutrients (lysine and proteins). We mainly replaced 100.9 g/kg of soy protein concentrated (Soycommil) by 80 g/kg of SDPP with small adjustment of other ingredients (barley, maize, soybean meal and synthetic amino acids) taking in consideration the different composition and amino acid profile between Soycomil and SDPP to provide close values for other nutritional components like fat or fiber and amino acid between both diets. We considered that the small adjustments increasing the content of 1.46% barley, 0,77% maize or 0.32% soybean meal in the SDPP diet or essential amino acids adjustment to balance the diets had little or no effect in the results observed in this study. Anyhow, following reviewer suggestion, we have included an extra sentence considering this in the discussion. Lines 128-129.
- line 173: what qRT-PCR stands for?
We apologize for the mistake of not describing it in the original version. This acronym stands for Real-Time quantitative PCR. Following reviewer suggestion, we clarified this acronym in the new version of the manuscript. Line 178.
- x axis label is hiding bar's label. why these three dots in the label?
Once again, we appreciated that reviewer made us aware of this mistake. We adjusted it in the revised version of the manuscript.
- Line 270: although this is true, the data cannot actually reflect the actual situation. there is no housekeeping gene normalization or accurate detection of genome copies.
The authors not fully understand the question from Reviewer 1. As commented in Material and Methods section we used the standardized qRT/PCR method approved by the OIEto analyze these samples. This method does not contain housekeeping gene normalization but is well accepted that differences in Ct values between samples correspond to difference in genome copies of the samples.
Following reviewer suggestion, we made a clarification in the revised version of the manuscript as follow “Viremia was found on d29pe (d12pse) in all animals except for one pig from the SDPP group (#5) and the average Ct values using the standardized qRT-PCR approved by the OIE in the SDPP group was numerically higher (indicating less ASFV genome copies) than in the CONVENTIONAL group.” Lines: 302-303
- Do the authors think that infecting animals by injection (not through contact) will show the same trend?
In our research group, we have routinely confirmed that the “in contact ASFV challenge” is as reproducible as the direct inoculation pf ASFV model. This is not a challenge sporadically performed in the laboratory (for example in Bosch-Camós et al., 2022), but a 100% efficient challenge model, reliable if keeping the optimal ratio of Trojans versus contact pigs. In fact, the experiments shown in this manuscript after second contact with 1:2 ratio, confirmed once again the reliable nature of a model that could be translated to any other laboratory, always keeping the mentioned ratio. Moreover, we consider that this is a more natural model of infection compared to the one in which we inject the virus intramuscularly.
The only difference between the in-contact challenge and the direct intramuscular inoculation (for example using the twenty lethal doses fifty 20LD50), is the kinetics of the infection. Thus, the onset of infection is faster for IM injected pigs with fever and ASFV being detectable from day four post-challenge. Conversely, in-contact challenge pigs develop similar clinical signs, including fever, viremia and shedding with 3 days of delay, reaching any how similar maximum virus titers and ASF compatible signs than those injected IM. As above mentioned, the in-contact challenge is 100% reproducible and as synchronic as the direct IM inoculation. Furthermore, we believe it mimics more closely the real situation found in nature. Following reviewer suggestion, we added a sentence in the discussion comparing both challenge models. Lines 356-364

Reviewer 2 Report
The manuscript by Blázquez et al reported that two groups of twelve weaned pigs each were fed with CONVENTIONAL or 8% SDPP enriched diets. After the second exposure, rectal temperature of conventionally fed contact pigs increased >40.5ËšC while fever was delayed in the SDPP contact pigs. PCR Ct values in blood, secretions and tissue samples were significantly lower (P<0.05) for CONVENTIONAL compared to SDPP contact pigs. Contact exposed pigs fed SDPP had delayed ASFV transmission and reduced virus load, likely by enhanced specific T-cell priming after the first ASFV-exposure.
Nevertheless, there are many Major Compulsory Revisions to be faced by the authors.
1. The molecular mechanisms that why the contact-exposed pigs fed SDPP showed lower virus load compared to conventional pigs have not been fully elucidated. The authors should focus on it and add more data in this manuscript.
2. The supplementary data are boring, they are primary data. The authors should further process the data.
3. In Figure 5, only showing the data of IFN-gamma is not enough. To make the conclusion in the manuscript more convincing, the authors must test other factors such as IFN-alpha and IFN-beta or cytokines. Furthermore, more time points are needed (only one point (11 dpc) is not convincing).
4. In Table 2, the authors should show it in a figure. Additionally, the main target organ of ASFV is the lung. However, no data regarding this organ are shown, which is very important to address the effects of SDPP. The authors must add the data.
5. To better demonstrate the effects of SDPP feeding, except for the testing in the paper, the authors should add the results of average daily gain, clinical signs and lesions of the lung and spleen.
Author Response
The reviewer #2 suggested an extensive editing of English language and style required, however, did not provide specific examples of why he/she is recommending Extensive English revision. Before submission, this manuscript has been extensively reviewed by our team and US native technical team.
In fact, the other reviewers for this manuscript and for the second manuscript (back-to-back submission) that we provide at the same time, did not indicated a concern related with the English edition.
We recommend the Reviewer #2 to provide specific examples that she/he consider need to be improve for English grammar in the revised version of the manuscript.
The manuscript by Blázquez et al reported that two groups of twelve weaned pigs each were fed with CONVENTIONAL or 8%SDPP enriched diets. After the second exposure, rectal temperature of conventionally fed contact pigs increased>40.5ËšC while fever was delayed in the SDPP contact pigs. PCR Ct values in blood, secretions and tissue samples were significantly lower (P<0.05) for CONVENTIONAL compared to SDPP contact pigs. Contact exposed pigs fed SDPP had delayed ASFV transmission and reduced virus load, likely by enhanced specific T-cell priming after the first ASFV-exposure.
Nevertheless, there are many Major Compulsory Revisions to be faced by the authors.
- The molecular mechanisms that why the contact-exposed pigs fed SDPP showed lower virus load compared to conventional pigs have not been fully elucidated. The authors should focus on it and add more data in this manuscript.
We thank you the Reviewer# 2 for his/her comments.
Regarding the first concern, we should be clear about the intention of the present manuscript and if needed softening our claims. We are far from being capable to provide a mechanistic explanation for the results observed. However, we believe that this is an exploratory work that provides interesting observations that deserve being published by itself, overall, when these effects seemed to be confirmed by the beneficial effects observed when feeding SDPP enriched diets during ASFV experimental vaccination (see accompanying manuscript 2).
We have added to the discussion (since we cannot conclude it) the following sentence (we hope the reviewer #2 consider convenient starting it with his/her own words): “The molecular mechanisms explaining why the contact-exposed pigs fed SDPP showed lower virus load compared to conventional pigs have not been elucidated. We believe that the accelerated ASFV specific T-cell response detected in this group of pigs after the first ASFV suboptimal encounter, might partially explain the protection afforded against the second challenge, this time using the optimal ratio of trojans”. Lines: 397-402 in the revised manuscript.
- The supplementary data are boring; they are primary data. The authors should further process the data.
The objective of providing the supplementary data was double, first to provide all set of information and secondly to have full transparency of the data obtained in this study (for those interested in specific details). We believe that adding this data (even after further processing) to the manuscript would contribute to boring readers, so we rather prefer to leave then as other reviewers found the information in this tale very complementary for understanding the whole information provided in this manuscript.
- In Figure 5, only showing the data of IFN-gamma is not enough. To make the conclusion in the manuscript more convincing, the authors must test other factors such as IFN-alpha and IFN-beta or cytokines. Furthermore, more time points are needed (only one point (11 dpc) is not convincing).
As far as we are aware, the mechanisms of protection are not totally elucidated for ASFV and might be multifactorial. Anyhow, and as described above (see answer to the first question addressed by the reviewer 1), we did not intend to provide a full mechanistic explanation to the observations performed.
This is in our understanding the first article providing evidence that dietary intervention might affect ASFV transmission and as such, we believe that we accomplished our objective.
Once learned the potential effect of SDPP on T cell responses, in the second article entitled “Feeding spray-dried porcine plasma to pigs improves the protection afforded by the African swine fever virus (ASFV) BA71∆CD2 vaccine prototype against experimental challenge with the pandemic ASFV. Study 2.” (accompanying this one), we provided a full set of data characterizing the direct effects of SDPP diet supplementation on the cytokines detected in sera (samples to study the effect of SDPP in the innate immune responses induced were taken).
- In Table 2, the authors should show it in a figure. Additionally, the main target organ of ASFV is the lung. However, no data regarding this organ are shown, which is very important to address the effects of SDPP. The authors must add the data.
Following the reviewer suggestion, we present the results of old table 2 as a new figure (Figure 2A).
Regarding the request for lung analysis, we respectfully disagree with the reviewer in this case. “ASFV causes a systemic infection in which the sites of primary replication are the monocytes and macrophages of the lymph nodes nearest to the point of entry” (Sánchez-Vizcaino et al., 2019). “Subsequently the virus spread via the blood and/or lymphatic system to sites of secondary replication: lymph nodes, bone marrow, spleen, lung, liver and kidney” (Sánchez-Vizcaino et al., 2019). These texts are extracted literally from the book Diseases of Swine, from the chapter on ASFV, and clearly indicate that lung is just one more tissue affected. Moreover, by our experience, lymphoid tissues are those that display more evident gross and microscopic lesions (Galindo-Cardiel et al., 2013. doi: 10.1016/j.virusres.2012.12.018) and harbor higher viral loads (Oh et al., 2022. doi: 10.3389/fvets.2022.978398) and, therefore, they were the ones used in this study. Certainly, we can accept that lung also harbor significant amounts of virus, but pulmonary information would not change obtained results based on previous comparative studies among tissues (Oh et al., 2022).
- To better demonstrate the effects of SDPP feeding, except for the testing in the paper, the authors should add the results of average daily gain, clinical signs and lesions of the lung and spleen.
We thank the reviewer for his/her constructive comments. However, ASF clinical signs are so dramatic, that average weight is never evaluated, and even less when working under BSL3 condition in an experimental set up with a reduced number of animals (it is not a clinical study and we have to fit with the 3Rs principles of the animal experimentation – reduction, refinement, and replacement).
On the other hand, as we highlighted in our previous answer, lungs were not considered to evaluate the viral load, since we had already five lymphoid tissues studied for this purpose. Regarding the spleen, a number of animals displayed splenomegaly; this information can be added but the authors consider that will not provide any additional information to the manuscript as the effect over the spleen and the other organs clearly suggest how the animals were affected.

Reviewer 3 Report
p2 L 66-68: authors claim that "Spray-dried plasma (SDP) .... that have biological activity independent of their nutritional value". but do not provide references demonstrating that specific biological activities are maintained in SDPP. References demonstrating biologic activities in SDPP should be provided.
p2 L 69-85 several references are provided that ascertain the beneficial effects of SDPP on growth characteristics. Of the 10 references provided, eight references were provided from two Spanish groups that also coauthored this article, and potential conflicts of interest are mentioned for authors who are employees of SDPP producers who also provided part of the funding.
The days of the experiment are labelled Dn (fig 2), dpc ( l.223, Table S3), or pse(l. 234). This is more confusing than helpful. The top is obtained at l. 293, which refers to day 25 in the text (which is 7days pse) but is 11dpc in Fig. 5. According to me 11dpc is eight days before day 0 of pse. Why is referring to a fig showing 11dpc data when speaking about the d25/7pse results?
L.254-56 some results are globalized ( rectal T°, viremia, rectal or nasal swab) to extract one information ( ASFV positivity) which is related to only one of the results (anal or rectal swabs).
L. 262 Based on delayed onset of clinical sign, lower viremia load, ASFV shedding, and ASFV genome copies authors claimed a reduced rate of in-pen transmission This assertion is quite surprising, as no transmission rate evaluation was provided. As shown in Table 2, all contact pigs turned ASFV positive, and ct was sometimes significantly lower for the SDPP groups. There seems to be confusion between the symptom evaluation, which according to the authors, is milder in the SDPP group, and the transmission rate. The clinical signs cannot be interpreted as an evaluation of transmission rate, as illustrated by the pig no.5 in the SDPP group, which displayed no variation in rectal temperature but was ASFV positive by qRT-PCR.
Furthermore, as shown in Table 3, there were indeed two pigs below ct 35 at day21 dpc but the ct values were 34, which is not a very convincing result to claim "a reduced rate of in-pen transmission" . Furthermore, by day 25dpc there were three ASFV-positive pigs in the nasal swabs of the conventional groups (ct 31, 31, and 32), but all the pigs were positive in the DSPP group (ct 29,31,27, 29, 26 30). Taking all the data together, I would be extremely careful before claiming a “reduced rates of in-pen transmission” which is not ascertained by the data.Fig 5 clearly illustrate that all SDPP pigs have been in contact to ASFV according to their ASFV-specific IFN-γ secreting T-cells positive results whereas only two pigs only are positive in the conventional group
l.288 sup table 6 is presented as anti ASFV antibodies results but contains elispot results and vice versa for sup table 7.
l. 323 the authors claim that nasal swab positive for ASFV was lower in SDPP contact group. The data provided in Table S3 show that: i) by day 4dpc one conventional contact pig turned positive but no SDPP contact pig; ii) by day7, five additional conventional contact pigs turned positive versus three SDPP contact pigs, but in this experiment one Trojan pig in the SDPP group remained ASFV negative. Under these conditions, the infectious challenge for conventional or SDPP contact pigs is not the same, which may indeed be related to the effect of SDPP, but adequate controls that would be infected with SDPP versus conventional pigs and infected conventional versus SDPP pigs are missing to provide any valid conclusion.
This discussion is astonishing.
The authors started by considerably minimizing the transmissibility of ASFV, especially via feed, and promoted feed supplementation with porcine plasma or SDPP. At any moment, the possibility of a risk of transmission with SDPP ( processing T is 80 °C and does not inactivate ASFV) is evoked. Then they follow this discussion by explaining that SDPP feeded pigs control the infection, cf l. 248: “two of the three trojan pigs in the SDPP group survived until the last day of the study with RT below 41ºC, even though these pigs displayed severe signs of disease.” Without any discussion about possible negative impact
All the discussion is turned toward promoting the use of SDPP, when in fact this work demonstrates an increased risk of dissemination of a deadly virus for the swine industry. Attenuation of ASF symptoms and a longer infection time, in fine, means an increased risk of ASFV-positive pigs reaching slaughtering houses and contamination of pork products, of which the plasma is used for SDPP, this scenario is completly ignored by the authors.
22 of the 42 references in the bibliography are signed by a least one coauthor of the article which is a lot to me.
Author Response
p2 L 66-68: authors claim that "Spray-dried plasma (SDP) .... that have biological activity independent of their nutritional value". but do not provide references demonstrating that specific biological activities are maintained in SDPP. References demonstrating biologic activities in SDPP should be provided.
Reply: There are more than 600 peer reviewed papers about the use of SDP in different applications and conditions. SDP has been one of the most researched ingredients use in pig nutrition and there are a significant number of scientific publications either in the target animal or animals models suggesting that plasma has a biological effect independently of the nutritional value, starting with the point that SDP works better under more stressing conditions of the animals. Following reviewer suggestion, we have added few references proving the biological activity of SDP in different animal models in the revised version of the manuscript.
p2 L 69-85 several references are provided that ascertain the beneficial effects of SDPP on growth characteristics. Of the 10 references provided, eight references were provided from two Spanish groups that also coauthored this article, and potential conflicts of interest are mentioned for authors who are employees of SDPP producers who also provided part of the funding.
Reply: As indicated in the manuscript, this study was funded by a SDPP producer following a previous study (Blázquez et al., 2020) in which the authors proved that feeding for 14 consecutive days commercial feed blended with liquid porcine plasma inoculated with ASFV from serum of infected pig at 10^4 and 10^5 TCID50/g was not infective in naïve pigs. Therefore, we were interested to understand the potential benefits of feeding SDPP to pigs challenged by contact with infected pigs. Nevertheless, the transparency of the role of each author and the conflicts of interest are clearly established in the Author contributions and Conflict of Interest sections.
Worldwide, excluding the case of China (for which the number of SDPP manufacturing plants is unknown), less than 15 companies with manufacturing plants in different countries of different geographical regions are involved in collecting and adding value to animal blood that contribute to the sustainability of the full meat industry. From these companies, unfortunately only one company has been consistently and continuously during more than 35 years investing resources working with leading universities and research centers to understand the full value of collecting blood products and potential applications either at human, animal and plant nutrition. Therefore, it is not surprising that most of the research publications available worldwide involved studies led during the last three decades by this company. However, all studies presented in this manuscript has been published in well-recognized, international scientific journal and therefore, been followed peer reviewing. Furthermore, most of these papers has been published in cooperation with very well recognized universities and research institutes that confirm the value of the finding obtained in most of them and preclude any bias that the reviewer is suggesting. We are surprised that the reviewer considers this situation as negative when in reality is rather a positive example of a company investing in applied research for the whole animal production industry. What is rather discouraging is that other companies have not been so involved in trying to understand the value of animal blood products in the different applications.
Nevertheless, following reviewer suggestion, we added a few more references in this section from publications unrelated with these two Spanish groups.
The days of the experiment are labelled Dn (fig 2), dpc ( l.223, Table S3), or pse(l. 234). This is more confusing than helpful. The top is obtained at l. 293, which refers to day 25 in the text (which is 7days pse) but is 11dpc in Fig. 5. According to me 11dpc is eight days before day 0 of pse. Why is referring to a fig showing 11dpc data when speaking about the d25/7pse results?
Reply: We thank you reviewer #3 for her/his comment. We agreed that probably the indication used in the manuscript can be confounded and therefore we decided to change this throughout the manuscript hoping clarifying this point.
L.254-56 some results are globalized (rectal T°, viremia, rectal or nasal swab) to extract one information (ASFV positivity) which is related to only one of the results (anal or rectal swabs).
Reply: We appreciate the comment from Reviewer #3. Certainly, this paragraph was not accurate in the submitted version and following reviewer recommendation, we changed it to the following paragraph:
“Viremia was detected in both groups at d29pe (d12pse). ASFV qPCR positive nasal swab was presented in all pigs for both groups from d23pe (d6pse) probably confirming the contact of these pigs with the second group of trojans. ASFV qPCR positive rectal swab was found in one pigs of the non-treated group on d23pe although at very low level (Ct 33); in contrast, the presence of ASFV virus in feces of pigs in the SDPP group was not found until d26pe (d9pse). Figure 4. Suppl. Table S3 and Suppl. Table S4” Lines 270-275 of the revised version
L. 262 Based on delayed onset of clinical sign, lower viremia load, ASFV shedding, and ASFV genome copies authors claimed a reduced rate of in-pen transmission This assertion is quite surprising, as no transmission rate evaluation was provided. As shown in Table 2, all contact pigs turned ASFV positive, and ct was sometimes significantly lower for the SDPP groups. There seems to be confusion between the symptom evaluation, which according to the authors, is milder in the SDPP group, and the transmission rate. The clinical signs cannot be interpreted as an evaluation of transmission rate, as illustrated by the pig no.5 in the SDPP group, which displayed no variation in rectal temperature but was ASFV positive by qRT-PCR. Furthermore, as shown in Table 3, there were indeed two pigs below ct 35 at day21 dpc but the ct values were 34, which is not a very convincing result to claim "a reduced rate of in-pen transmission" . Furthermore, by day 25dpc there were three ASFV-positive pigs in the nasal swabs of the conventional groups (ct 31, 31, and 32), but all the pigs were positive in the DSPP group (ct 29,31,27, 29, 26 30). Taking all the data together, I would be extremely careful before claiming a “reduced rates of in-pen transmission” which is not ascertained by the data. Fig 5 clearly illustrates that all SDPP pigs have been in contact to ASFV according to their ASFV-specific IFN-γ secreting T-cells positive results whereas only two pigs only are positive in the conventional group.
Reply: We may agree with Reviewer #3 that according to the conditions for the study, it is difficult to conclude a reduced transmission rate because it was difficult to be evaluated. However, from the fact that two of the three second inoculated trojans survived until the end in the SDPP group, probably spreading viruses for additional 4 days, it was expected that contact pigs with these two surviving trojans to be rapidly infected compared with the control group, and, from the nasal swab analysis, all pigs in the SDPP were in contact with the trojans infected pigs as can be observed on the results at d23pe, d26pe8 and d29pe31, however, these animals did not excrete virus in feces until d26pe. However, in the control group, with less time of animals exposed to infected trojans, some animals contain the virus genome in feces from d19pe onwards. To us, this may indicate a lower rate of transmission. Nevertheless, following the suggestion from Reviewer #4, we changed the paragraph to “…which may indicate reduced rate of in-pen transmission”.
l.288 sup table 6 is presented as anti ASFV antibodies results but contains elispot results and vice versa for sup table 7.
Reply: We appreciate that the reviewer has seen this error in the numbering of the tables, and we have corrected it in the new revision of the manuscript. Lines 433-435
l. 323 the authors claim that nasal swab positive for ASFV was lower in SDPP contact group. The data provided in Table S3 show that: i) by day 4dpc one conventional contact pig turned positive but no SDPP contact pig; ii) by day7, five additional conventional contact pigs turned positive versus three SDPP contact pigs, but in this experiment one Trojan pig in the SDPP group remained ASFV negative. Under these conditions, the infectious challenge for conventional or SDPP contact pigs is not the same, which may indeed be related to the effect of SDPP, but adequate controls that would be infected with SDPP versus conventional pigs and infected conventional versus SDPP pigs are missing to provide any valid conclusion.
Reply: We agree with reviewer comment that maybe the effect of less nasal swab positive samples after the first exposure can be related with one trojan pig in this group excreting less ASFV compared with the control group. However, that pig died at the same time than the trojans in the control group and is unclear if the diet made this trojan pig to delay the ASFV clinical signs appearance. Therefore, following reviewer recommendation, we changed this paragraph in the new version of the manuscript:
“The fact that positive nasal swab samples were detected in a lower number of contact pigs in SDPP versus the CONVENTIONAL group, can be related with either less virus excretion from the trojan pigs fed SDPP diet during first exposure or related to our original hypothesis of innate response improvement by SDPP feeding [18,29].” Lines 368-371.
This discussion is astonishing.
Reply: We respectfully disagree with Reviewer #3 comment. During the discussion we have tried our best to explain the results obtained in this study and connect to what is known from other studies with SDPP supplementation. As far as we are aware, the mechanisms of protection are not totally elucidated for ASFV and might be multifactorial. Anyhow, we did not intend to provide a full mechanistic explanation to the observations performed. This is, to our understanding, the first article providing proof-of-concept evidence that dietary intervention might affect ASFV transmission and, as such, we believe we accomplished our objective.
The authors started by considerably minimizing the transmissibility of ASFV, especially via feed, and promoted feed supplementation with porcine plasma or SDPP. At any moment, the possibility of a risk of transmission with SDPP (processing T is 80 °C and does not inactivate ASFV) is evoked.
Reply: We respectfully disagree with reviewer #3 comment. During the first part of discussion, we reviewed the fact that, during the first exposure time, none of the pigs (20 for both groups) were infected by contact with trojans pigs and we referred to reference 30 (Schulz et al., 2019) to support the statement that ASFV is less transmissible compared with other pig viruses. We also related this observation with a previous study in which we found that feeding naïve pigs with commercial feed blended with liquid plasma inoculated with high doses of ASFV (104.3 or 105.0 TCID50/g) was not infective when provided for 14 consecutive days, when other feeding studies with no SDPP have been reported to be infective, and therefore, as indicated early, this observation was the reason for conducting this study.
As the focus of the study was to evaluate the nutritional intervention of feeding SDPP to pigs challenged with ASFV by contact, we did not include references related to the safety of SDPP because was not the objective of this study. There is the assumption that blood products are collected always from areas free from the disease. However, the safety of the manufacturing process of SDPP in front of multiple diseases of concern in the swine industry has been extensively tackled in numerous peer review papers, including the safety of the manufacturing process against ASFV. Probably SDPP is one of the ingredients that has been more investigated in relation of the safety for different pathogens affecting farm animals due to the perceived risk of pathogen transmission. In the below table there is a short list of safety studies of SDPP in front a diversity of viruses affecting the pig industry. One of these papers (Blázquez et al., 2021) demonstrated that the spray-drying process of SDPP inactivated between 2.1 to 4.2 ASFV TCID50/g inoculated in liquid plasma and dried in laboratory spray-drier and submitted to different residence time inside the drier (from 1 sec in lab drier to 90 second as happen in industrial driers). A new publication of the combination of spray-drying with storage the dried product for 14 days at either 4ºC or 20ºC demonstrating full inactivation >5.18 TCID50/g has been recently submitted to a peer reviewed journal and hopefully will be available soon. Furthermore, in 2021, the EFSA published the risk assessment of different ingredients matrices for ASFV transmission, and blood products were positioned in the lower risk of transmission compared with the rest of ingredients (from vegetal or animal origin) as results of the recognition of the strong regulatory control by the competent authorities and the multiple hurdle steps involved in the manufacturing process (EFSA Journal 2021;19(4):6558 doi: 10.2903/j.efsa.2021.6558).
|
Polo, J., J. D. Quigley, L. E. Russell, J. M. Campbell, J. Pujols, and P. D. Lukert. 2005. Efficacy of spray-drying to reduce infectivity of pseudorabies and porcine reproductive and respiratory syndrome (PRRS) viruses and seroconversion in pigs fed diets containing spray-dried animal plasma. J. Anim. Sci. 83:1933-1938. |
|
Pujols, J., R. Rosell, L. Russell, J. Campbell, J. Crenshaw, E. Weaver, C. Rodríguez, J. Ródenas, and J. Polo. 2007. Inactivation of swine vesicular disease virus in porcine plasma by spray-drying. Proc. Amer. Assoc. Swine Vet. p. 281-283. |
|
Pujols, J., S. Lopez-Soria, J. Segalés, M. Fort, M. Sibila, R. Rosell, D. Solanes, L. Russell, J. Campbell, J. Crenshaw, E. Weaver, and J. Polo. 2008. Lack of transmission of porcine circovirus type 2 to weanling pigs by feeding them spray-dried porcine plasma. Vet. Rec. 163:536-538 |
|
Pujols, J., C. Lorca-Oró, I. Díaz, L. E. Russell, J. M. Campbell, J. D. Crenshaw, J. Polo, E. Mateu, and J. Segalés. 2011. Commercial spray-dried porcine plasma does not transmit porcine circovirus type 2 in weaned pigs challenged with porcine reproductive and respiratory syndrome virus. Vet. J. 190:e16-e20. |
|
Shen, H. G., S. Schalk, P. G. Halbur, J. M. Campbell, L. E. Russell, and T. Opriessnig. 2011. Commercially produced spray-dried porcine plasma contains increased concentrations of porcine circovirus type 2 DNA but does not transmit porcine circovirus type 2 when fed to naïve pigs. J. Anim. Sci. 89:1930-1938. |
|
Polo, J., T. Opriessnig, K. C. O’Neill, C. Rodríguez, L. E. Russell, J. M. Campbell, J. Crenshaw, J. Segalés, and J. Pujols. 2013. Neutralizing antibodies against porcine circovirus type 2 in liquid pooled plasma contribute to the biosafety of commercially manufactured spray-dried porcine plasma. J. Anim. Sci. 91:2192-2198. |
|
Pujols J and J. Segalés. 2014. Survivability of porcine epidemic diarrhea virus (PEDV) in bovine plasma submitted to spray drying processing and held at different time by temperature storage conditions. Vet Microbiol 174:427-432 |
|
Dee S., T. Clement, A. Schelkopf, J. Nerem, D. Knudsen and J. Christopher-Hennings. 2014. An evaluation of contaminated complete feed as a vehicle for porcine epidemic diarrhea virus infection of naïve pigs following consumption via natural feeding behavior: proof of concept. BMC Vet Res. 10:176-184. |
|
Opriessnig, T., C. T. Xiao, P. F. Gerber, J. Zhang, and P. G. Halbur. 2014. Porcine epidemic diarrhea virus RNA present in commercial spray-dried porcine plasma is not infectious to naïve pigs. Plos One. 9(8):e104766. doi:10.1371/journal.pone.0104766. |
|
Gerber, P. F., C. T. Xiao, Q. Chen, J. Zhang, P. G. Halbur, and T. Opriessnig. 2014. The spray-drying process is sufficient to inactivate infectious porcine epidemic diarrhea virus in plasma. Vet. Microbiol. 174:86-92. |
|
Pujols J., C. Rodríguez, N. Navarro, S. Pina-Pedrero, J.M. Campbell, J. Crenshaw, and J. Polo. 2015. No transmission of hepatitis E virus in pigs fed diets containing commercial spray-dried porcine plasma: a retrospective study of samples from several swine trials. Virology J 2014, 11:232. doi: 10.1186/s12985-014-0232-x. |
|
Duffy MA, Q. Chen, J. Zhang, PG Halbur and T. Opriessnig. 2018. Impact of dietary spray-dried bovine plasma addition on pigs infected with porcine epidemic diarrhea virus. Transl. Anim. Sci. 2018.2:349–357 |
|
Kalmar ID, AB Cay, and M Tignon. 2018. Sensitivity of African swine fever virus (ASFV) to heat, alkalinity and peroxide treatment in presence or absence of porcine plasma. Veterinary Microbiology 219: 144–149. |
|
Blazquez E, C Rodrıguez, J Rodenas, N Saborido, M Sola-Gines, A Perez de Rozas, JM Campbell, J Segales, J Pujols and J Polo. 2018. Combined effects of spray-drying conditions and postdrying storage time and temperature on Salmonella choleraesuis and Salmonella typhimurium survival when inoculated in liquid porcine plasma. Letters in Applied Microbiology. doi:10.1111/lam.13017 |
|
Russell, L. E., J. Polo, D. Meeker. 2020. The Canadian 2014 porcine epidemic diarrhea virus outbreak: Important risk factors that were not considered in the epidemiological investigation could change the conclusions. Transbound and Emerg Dis. 2020;00:1-12. doi: 10.1111/tbed.13496 |
|
Blázquez, E., J. Pujols, J. Segalés, N. Navarro, C. Rodríguez, J. Ródenas, J. Polo. Inactivation of African swine fever virus inoculated in liquid plasma by spray drying and storage for 14 days at 4ºC or 20ºC. Submitted article. |
Then they follow this discussion by explaining that SDPP feeded pigs control the infection, cf l. 248: “two of the three trojan pigs in the SDPP group survived until the last day of the study with RT below 41ºC, even though these pigs displayed severe signs of disease.” Without any discussion about possible negative impact All the discussion is turned toward promoting the use of SDPP, when in fact this work demonstrates an increased risk of dissemination of a deadly virus for the swine industry. Attenuation of ASF symptoms and a longer infection time, in fine, means an increased risk of ASFV-positive pigs reaching slaughtering houses and contamination of pork products, of which the plasma is used for SDPP, this scenario is completely ignored by the authors.
Reply: As indicated in previous comment, the intention of this study was to evaluate if the nutritional intervention with SDPP in the diet was able to modulate the immune response of the animals when in contact with infected animals and reduce transmission and severity of the disease clinical signs. The authors were really surprised to find that two of the three IM inoculated trojans survived for 12 days in the SDPP group and we considered that this can be an interesting observation of how a functional ingredient can interact with the disease development and attenuate the speed of the disease. However, intramuscular injected pigs cannot be considered a natural model of infection; in agreement with the reviewer, probably this IM inoculated pigs that survived longer is negative for the spread of the disease, but under natural and farm conditions, the main way of virus transmission is by in contact with infected pigs. Therefore, we used this more natural infection model, and in such model the nutritional intervention resulted in ameliorating the outcome of the ASFV infection. In fact, this is also what we found in the previous publication mentioned before of feeding liquid ASFV inoculated plasma to naïve pigs without infecting the animals and the following study (back-to-back submission with this manuscript) in which the nutritional intervention with SDPP in the diet improved the efficacy of a prototype live attenuated ASFV vaccine.
22 of the 42 references in the bibliography are signed by a least one coauthor of the article which is a lot to me.
Reply: We referred to the reviewer to the previous comment to one of her/his first questions.

Reviewer 4 Report
Blázquez et al. present an interesting study on the effect of spray-dried porcine plasma on direct infection and in contact transmission of African swine fever. The results show a clear effect on both, but the data does completely support the authors interpretations. The manuscript is difficult to follow in sections due to the way the results are presented and the IFN ELISpot results need further controls and explanation. I think if the following modifications are made to the manuscript then the paper will make a substantial contribution to our understanding of how diet can effect the outcome of infection after a very important disease of domestic swine.
The authors variously use dpc, day and dp2e to refer to the time points in their study. Although I understand why this gets very confusing at times. It would be helpful if one term could be used through out, or that the graphs and tables refer to the different terms so that when the reader refers back to the figures they can quickly work out what the text is referring to.
Line 204. Which MOI was used? Surely there wasn't a ten-fold difference in the amount of virus used to stimulate the cells from the different pigs in Figure 5? If so then this should clearly be stated in the methods and figure. The nature of the mock mentioned in the legend for Figure 5 needs stating as does how the virus was prepared.
Line 229. Please include room conditions (temp, humidity, air changes) and cleaning regime employed. As the first transmission experiment failed this is important information!
Line 251. Although the length of time the donor pigs were in contact with the other animals was greater for the SDPP pigs the data suggests viraemia and shedding was lower which may also explain the differences rather than the feed itself. The data in Supplementary Figure 3 should be incorporated into the main body of the manuscript, preferably as a graphs.
Line 289. The presence of systemic cellular immune responses (T-cells in the blood) in the absence of any evidence of viral replication is very difficult to explain. The variable use of MOI suggested in the methods is concerning and makes interpretation of the results very difficult. The legend suggests that mock subtraction has employed, however I think this data would need to be shown. Furthermore if the mock is not a true mock inoculum (i.e. uninfected material prepared from the same cells as the virus) then data from pre-exposure cells needs to be included to confirm these results. The authors mention that SDPP induced higher cytokines and the results in the companion paper support this, therefore the source of the IFN after stimulation could be cells other than T-cells. In order to support the authors conclusions that an ASFV-specific T-cell response has been induced needs flow cytometry analysis. This is a lot of work and if the data is not available I think Figure 6 could be removed from the manuscript without seriously impacting the interesting observations of the effect of the SDPP diet on ASFV infection and transmission.
Author Response
Blázquez et al. present an interesting study on the effect of spray-dried porcine plasma on direct infection and in contact transmission of African swine fever. The results show a clear effect on both, but the data does completely support the authors interpretations. The manuscript is difficult to follow in sections due to the way the results are presented and the IFN ELISpot results need further controls and explanation. I think if the following modifications are made to the manuscript then the paper will make a substantial contribution to our understanding of how diet can affect the outcome of infection after a very important disease of domestic swine.
The authors variously use dpc, day and dp2e to refer to the timepoints in their study. Although I understand why this gets very confusing at times. It would be helpful if one term could be used throughout, or that the graphs and tables refer to the different terms so that when the reader refers back to the figures, they can quickly work out what the text is referring to.
Reply: We thank reviewer #4 for her/his positive and constructive review. We have changed the way we use to refer to the timepoints in the study. Hopefully the new version of the manuscript will be easy to follow by readers.
Line 204. Which MOI was used? Surely there wasn't a ten-fold difference in the amount of virus used to stimulate the cells from the different pigs in Figure 5? If so then this should clearly be stated in the methods and figure. The nature of the mock mentioned in the legend for Figure 5 needs stating as does how the virus was prepared.
Reply: We appreciated the comments received from reviewer #4 for clarification on the amount of MOI used in the study and improve details of the Elispot technique used in our study. Following reviewer suggestion, we modify the description of the technique in the material and methods section.
“Peripheral blood mononuclear cells (PBMCs) were purified from EDTA blood samples by density-gradient centrifugation with Histopaque 1077 (Sigma-Aldrich, Missouri, USA). Cells were frozen at -80ºC but the viability was tested to be ≥ 90% after thawing before conducting the analysis. Commercial antibody Porcine IFN-γ P2G10 and biotin P2C11, from BD Biosciences (Pharmingen, California, USA) at 5 µg/mL was used to quantify by ELISPOT assay the number of IFNγ secreting cells. PBMC were stimulated against ASF Georgia/01 strain virus at a MOI 0.2 to measure specific T-cell responses by ELISPOT in real time with fresh cells [35]. Phytohemagglutinin (PHA; Roche Diagnostics, Barcelona, ​​Spain) was used as a positive control and the mock was carried out using only cells and RPMI medium as a negative control. Plates were revealed using detection antibody from BD Pharmingen (BD Biosciences, California, USA), Strep-HRP from Life Technologies (California, USA) and TMB substrate for ELISPOT assay (MABTECH, Stockholm, Sweden). ELISPOTs were read under a magnifying glass. The value obtained for each animal was obtained subtracting the corresponding values of mock-stimulated cells”. Lines 209-222 of the revised version of the manuscript.
We also modify the legend of Figure 6 (old figure 5) in the new version of the manuscript.
“Figure 6. ASFV-specific IFN-γ secreting T-cells analyzed by ELISPOT at 9 days after first ASFV encounter (d9pe). PBMC from d9pe were stimulated in vitro with Georgia 2007/1 at MOI 0.2 and the number of ASFV-specific IFN-γ-secreting cells was quantified by ELISPOT. Values shown are individual values subtracting the corresponding values of mock-stimulated cells. C-#: Animals in CONVENTIONAL diet; SDPP-#: animals in spray-dried porcine plasma (SDPP) diet.” Lines 329-333.
Line 229. Please include room conditions (temp, humidity, air changes) and cleaning regime employed. As the first transmission experiment failed this is important information!.
Reply: The experimental conditions of the study have been added to the new version of the manuscript as following “The rooms contain slatted floor and the environmental conditions for both rooms were set at 22±2°C and relative humidity of 60±5%. The air renewal was established to be 12 times/hour. The feed was provided each morning between 7:30-9:30 am.” This information has now been included in the revised manuscript (Lines: 131-134).
Line 251. Although the length of time the donor pigs were in contact with the other animals was greater for the SDPP pigs the data suggests viraemia and shedding was lower which may also explain the differences rather than the feed itself. The data in Supplementary Figure 3 should be incorporated into the main body of the manuscript, preferably as a graph.
Reply: We again thank the reviewer for her/his comment. Being true that this might be an explanation, we rather believe that both effects are due to the SDPP diet supplementation, confirmed with the beneficial effects observed when feeding pigs with SDPP-enriched diets during ASFV vaccination (see back-to-back submitted manuscript 2).
We have taken the freedom to use her/his own words to include a new sentence in the discussion section.
“Although the length of time the donor pigs were in contact with the other animals was greater for the SDPP pigs, the data suggests viraemia and shedding was lower which may also contribute to explain the differences observed, rather than the feed itself. Further work should be performed in the future with a larger group of animals to confirm these observations. However, the accelerated specific T-cell responses observed in SDPP-fed animals upon the first encounter with the ASFV, the lower ASFV loads found in the same group after the second challenge, together with the beneficial effect that SPPP seems to exert upon experimental ASFV vaccination [49] (back-to-back submitted manuscript) would support the hypothesis that nutritional interventions may be useful to help ameliorating the impact of ASF in endemic regions” Lines 414-423.
As suggested by reviewer 4, old Supplementary Table 3. Has been incorporated into the main body of the manuscript as graphs.
Line 289. The presence of systemic cellular immune responses (T-cells in the blood) in the absence of any evidence of viral replication is very difficult to explain. The variable use of MOI suggested in the methods is concerning and makes interpretation of the results very difficult. The legend suggests that mock subtraction has employed, however I think this data would need to be shown. Furthermore, if the mock is not a true mock inoculum (i.e. uninfected material prepared from the same cells as the virus) then data from pre-exposure cells needs to be included to confirm these results. The authors mention that SDPP induced higher cytokines and the results in the companion paper support this, therefore the source of the IFN after stimulation could be cells other than T-cells. In order to support the authors conclusions that an ASFV-specific T-cell response has been induced needs flow cytometry analysis. This is a lot of work and if the data is not available I think Figure 6 could be removed from the manuscript without seriously impacting the interesting observations of the effect of the SDPP diet on ASFV infection and transmission.
Reply: Understanding the concern of the Reviewer #4, this manuscript does not intend to understand the molecular mechanisms behind the protection afforded. However, we believe this data deserves being maintained for several reasons. The first one, because we do believe it is a real evidence of specific T-cell priming; second, because we have confirmed that after experimental vaccination with live attenuated vaccine prototype, there is an improvement on the specific T cell induction in the SDPP fed animals, without any evidence of detectable vaccine virus in the blood (manuscript 2 submitted back-to-back); and third, because the other reviewers apparently agree, based on their comments, that this is an important piece information.

Round 2
Reviewer 1 Report
The authors have addressed all comments properly. The manuscript has improved significantly
Author Response
The authors have addressed all comments properly. The manuscript has improved significantly.
The authors really appreciate the comment from Reviewer #1 and all his/her help with the suggestions to improve the quality of the final manuscript. Thanks again for all his/her contribution to the final version

Reviewer 4 Report
The changes the authors have answered all of my queries except for the final one about priming T-cell responses. The data presented doesn't prove that the presence of ASFV specific T-cell responses they would need to use flow cytometry to do this, or to perform ELIspot on fractionated cells rather than whole PBMCs. The authors would need to demonstrate an absence of response to the virus preparation prior to the transmission (i.e. day 0) and an absence of a response to an appropriate control (i.e. uninfected cells harvested in the same way the virus inoculum was prepared). As the authors mention the SDPP diet led to increased levels of circulating cytokines and therefore the observed enhanced response to ASFV may not be specific. As I mentioned in my initial report, removing this from the manuscript would not affect the other conclusions.
Author Response
The changes the authors have answered all of my queries except for the final one about priming T-cell responses. The data presented doesn't prove that the presence of ASFV specific T-cell responses they would need to use flow cytometry to do this, or to perform ELIspot on fractionated cells rather than whole PBMCs. The authors would need to demonstrate an absence of response to the virus preparation prior to the transmission (i.e. day 0) and an absence of a response to an appropriate control (i.e. uninfected cells harvested in the same way the virus inoculum was prepared). As the authors mention the SDPP diet led to increased levels of circulating cytokines and therefore the observed enhanced response to ASFV may not be specific. As I mentioned in my initial report, removing this from the manuscript would not affect the other conclusions.
We apologize with the reviewer. The issue addressed has now been clarified in the text. We understand that results from the ELISpot assay performed with PBMC might not be necessarily linked to virus-specific T cells. However, our group has long experience on the quantification of ASFV-specific T cell responses, most of the times using PBMC from infected and/or vaccinated pigs as well as in non-infected/vaccinated animals and mock-stimulated cells as negative controls (Monteagudo et al,2018, J. Virol. 91:e01058-17; Netherton et al., 2019, Front Immunol.10:1318; Zhang et al., 2020, Sci. Rep. 10:1 10; Sun et al., 2021, Viruses 13(11):2264; Bosch-Camós et al., 2022. Plos Pathog. 18(11):e1010931). Furthermore, using the same virus stock as stimulus from the present study, we have recently performed Flow cytometry intracellular staining assays to quantify IFNG-producing ASFV-specific T cells, and demonstrated that the results are consistent with the ELISPOT assay. In any case, since the reviewer is correct when pointing out that in the present study, we do not have non-infected pigs as controls, we have now modified the text to avoid the use of the term “ASFV-specific T cells”. We use instead the term “ASFV-responding” cells, as we understand this is correct since we did use RPMI medium as negative control in the ELISPOT assay, as mentioned in the Material and Methods section (lines 216 to 222).
The authors prefer to keep this information in the manuscript for the reasons explained above and also because other reviewers agreed, based on their comments, that this can be a very useful information to keep in the final version.
The authors appreciate the insistence of the Reviewer for clarifying the results obtained with the ELISPOT.
